# Quantitative functional renormalization for three-dimensional quantum Heisenberg models

Nils Niggemann[1,2]*, Johannes Reuther[1,2] and Björn Sbierski[3,4]

**1** Dahlem Center for Complex Quantum Systems and Institut für Theoretische Physik, Freie Universität Berlin, Arnimallee 14, 14195 Berlin, Germany
**2** Helmholtz-Zentrum für Materialien und Energie, Hahn-Meitner-Platz 1, 14109 Berlin, Germany
**3** Department of Physics and Arnold Sommerfeld Center for Theoretical Physics, Ludwig-Maximilians-Universität München, Theresienstr. 37, 80333 Munich, Germany
**4** Munich Center for Quantum Science and Technology (MCQST), 80799 Munich, Germany
* nils.niggemann@fu-berlin.de

January 11, 2022

## Abstract

We employ a recently developed variant of the functional renormalization group method for spin systems, the so-called pseudo Majorana functional renormalization group, to investigate three-dimensional spin-1/2 Heisenberg models at finite temperatures. We study unfrustrated and frustrated Heisenberg systems on the simple cubic and pyrochlore lattices. Comparing our results with other quantum many-body techniques, we demonstrate a high quantitative accuracy of our method. Particularly, for the unfrustrated simple cubic lattice antiferromagnet ordering temperatures obtained from finite-size scaling of one-loop data deviate from error controlled quantum Monte Carlo results by $\sim 5\%$ and we further confirm the established values for the critical exponent $\nu$ and the anomalous dimension $\eta$. As the PM-FRG yields results in good agreement with QMC, but remains applicable when the system is frustrated, we next treat the pyrochlore Heisenberg antiferromagnet as a paradigmatic magnetically disordered system and find nearly perfect agreement of our two-loop static homogeneous susceptibility with other methods. We further investigate the broadening of pinch points in the spin structure factor as a result of quantum and thermal fluctuations and confirm a finite width in the extrapolated limit $T \rightarrow 0$. While extensions towards higher loop orders $\ell$ seem to systematically improve our approach for magnetically disordered systems we also discuss subtleties when increasing $\ell$ in the presence of magnetic order. Overall, the pseudo Majorana functional renormalization group is established as a powerful many-body technique in quantum magnetism with a wealth of possible future applications.

# 1  Introduction

A wide spectrum of magnetic phenomena occurs in systems described by a Heisenberg model
[1] in which spin-1/2 operators $\boldsymbol{S}_i$ located on lattice sites $i$ are coupled via isotropic exchange
interactions $J_{ij}$,

$$H = \sum_{i<j} J_{ij} \boldsymbol{S}_i \boldsymbol{S}_j. \tag{1}$$

In spite of the apparent simplicity of Eq. (1), the calculation of measurable quantities remains a
notoriously difficult problem, particularly in the most realistic case of three spatial dimensions.
Numerical methods, while indispensable and of steadily increasing power, usually either suffer
from an intrinsic bias, are limited in the quantitative accuracy of their predictions or are unfeasible
for the treatment of generic three-dimensional (3D) systems.

Besides more established approaches such as quantum Monte Carlo (QMC) [2], exact diago-
nalization [3], and density-matrix renormalization group (DMRG) [4], new concepts like the func-
tional renormalization group [5, 6] are currently on the rise for spin systems, owing to their flex-
ibility and applicability to even complex coupling scenarios. While it is now possible to directly
treat the RG flow of spin-vertex functions [7], more established variants represent spin operators
in terms of auxiliary fermions. The pseudofermion functional renormalization group (PFFRG)
method [8–12] is particularly strong in calculating ground state spin correlations while a more re-
cent variant, the pseudo *Majorana* functional renormalization group (PMFRG) approach [13] can
even handle combined effects of quantum and thermal fluctuations. On the other hand, these meth-
ods are sometimes associated with the weaknesses that ($i$) they are in no simple way endowed with
a parameter that systematically controls the accuracy and ($ii$) rigorous benchmark tests with other
methods are rarely possible. The recent application of multiloop FRG extension [14, 15] to the
PFFRG [16, 17] has made an important step forward concerning ($i$) by systematically increasing
the loop order $\ell$ of diagrammatic contributions to the vertex flow.

In this work, we tackle ($ii$) by exploiting the PMFRG's capability of treating finite tempera-
tures which opens up a plethora of further applications and opportunities for benchmarking. We
apply the PMFRG to two types of models; the first ones are unfrustrated 3D systems such as
the nearest-neigbor simple cubic lattice antiferromagnet where one expects a finite temperature
transition to a magnetically ordered state. Details of these second-order phase transitions such as
the critical temperature and -exponents are well studied from QMC [18] which treats unfrustrated
models in a completely unbiased and error-controlled way. For the PMFRG, probing universal

finite-size scaling [19] behaviors provides an optimal testbed and allows us to demonstrate its beyond-mean-field character in a quantitative and rigorous way. Overall, we find QMC results very well reproduced, which concerns the values of critical temperatures $T_c$, the critical exponent for the correlation length $\nu$ which we confirm via a scaling collapse, and the anomalous dimension $\eta$. An interesting byproduct of our results is the insight that the system size parameter $L$ which in PMFRG limits the range of spin-correlations can be used for finite-size scalings in a similar way as the box-size in QMC.

The surprisingly accurate PMFRG results for magnetically ordered systems motivate us to move on to a second type of models where frustration effects are strong enough to suppress magnetic long-range order at low temperatures. As a paradigmatic geometrically frustrated system, we investigate the nearest neighbor pyrochlore Heisenberg antiferromagnet [20], which is known for its rich phenomenology related to spin ice systems. In this context, we also partially tackle the aforementioned point $(i)$ by extending the PMFRG with two-loop ($\ell = 2$) corrections but leave even higher loop orders for future work. Since QMC is no longer applicable to such systems due to the sign problem, possibilities for benchmark checks become rarer. Whenever comparisons are possible, e.g. for the homogeneous susceptibility of the pyrochlore antiferromagnet, our results show remarkable agreement with other numerical approaches. We also investigate long-standing open problems in the field of quantum magnetism such as the fate of pinch point singularities [21] in the pyrochlore Heisenberg antiferromagnet and the possibility of a magnetically disordered low-temperature phase on the simple cubic lattice with second neighbor interactions [22–27].

Overall, our results demonstrate that for unfrustrated systems PMFRG are in quantitative agreement with QMC, but has the additional advantage of being also applicable to frustrated systems where the high accuracy is expected to persist. Therefore, besides the results presented below, we believe that our work has important implications for future investigations of quantum magnetic systems, establishing the PMFRG as a flexible and powerful method, applicable to both unfrustrated and frustrated systems. However, it is also worth emphasizing that this work does not conclude the development of the PMFRG. Particularly, we expect that multiloop extensions with $\ell \geq 3$ yield further important insights into quantum magnets and may enable the exploration of lower temperature regimes which are not reachable within our current implementation.

The paper is structured as follows: After a brief review of the PMFRG's formalism in Section 2, we study magnetic phase transitions on the simple cubic lattice using rigorous finite-size scaling laws in Section 3. Subsequently, we turn to the strongly frustrated nearest-neighbor pyrochlore model in Section 4 and investigate the static $q = 0$ spin susceptibility as well as pinch-point-like features in the spin structure factor. Here, we also discuss improvements to the susceptibility introduced by two-loop corrections as well as measurements of the energy per site and the specific heat capacity. Finally, we discuss the effects of two-loop corrections more broadly in Section 5 and summarize our results in Section 6. Appendices are devoted to more technical aspects.

## 2 Pseudo-Majorana functional renormalization group

In this section, we briefly sketch the PMFRG approach. For a more in-depth introduction, we refer the interested reader to Ref. [13] and to our Appendices. Using the $SO(3)$ Majorana representation of spins $S_i^x = -i\eta_i^y \eta_i^z$, $S_i^y = -i\eta_i^z \eta_i^x$, $S_i^z = -i\eta_i^x \eta_i^y$ with $\{\eta_i^\alpha, \eta_j^\beta\} = \delta_{ij}\delta^{\alpha\beta}$ [28, 29], the PMFRG can be applied to Heisenberg systems [Eq. (1)]. At its core lies the solution of a (truncated) system of functional renormalization group flow equations [5, 30] which are differential equations for the irreducible vertices as functions of a cutoff parameter $\Lambda$. In the present case, a smooth cutoff is chosen which modifies the bare Green's function as $G^{0,\Lambda}(\omega_n) = \Theta^\Lambda(\omega_n)G^0(\omega_n)$ with $\Theta^\Lambda(\omega_n) = \frac{\omega_n^2}{\omega_n^2 + \Lambda^2}$ where $\omega_n$ is a Matsubara frequency. However, we note that we found neg-

ligible dependence of our results upon the choice of the cutoff function $\Theta^\Lambda(\omega_n)$. Grouping site-, flavor- and frequency indices together as $1 \equiv (i_1, \alpha_1, i\omega_{n_1})$, the one-loop flow equations for the interaction correction to the free energy $F_{\text{int}}$, the self energy $\Sigma$ and the four-point vertex $\Gamma$ are

$$\frac{d}{d\Lambda} F_{\text{int}}^\Lambda = \frac{1}{2} \text{Tr} \left[ \dot{G}^\Lambda G^{0,\Lambda} \left[ G^\Lambda \right]^{-1} \Sigma^\Lambda \right], \tag{2a}$$

$$\frac{d}{d\Lambda} \Sigma_{1,2}^\Lambda = -\frac{1}{2} \sum_{1',2'} \dot{G}_{1',2'}^\Lambda \Gamma_{1'2',1,2}^\Lambda, \tag{2b}$$

$$\frac{d}{d\Lambda} \Gamma_{1,2,3,4}^\Lambda = X_{1,2|3,4}^\Lambda - X_{1,3|2,4}^\Lambda + X_{1,4|2,3}^\Lambda, \tag{2c}$$

$$X_{1,2|3,4}^\Lambda = \sum_{1',\dots,4'} \dot{G}_{1',2'}^\Lambda G_{3',4'}^\Lambda \Gamma_{1,2,1',3'}^\Lambda \Gamma_{2',4',3,4}^\Lambda. \tag{2d}$$

Here, $G^\Lambda$ is the cutoff modified version of the two-point Green's function $G_{1,2} = \langle \eta_1 \eta_2 \rangle$ and $\dot{G}_{1,2}^\Lambda$ the single-scale propagator. In Eq. (2d), Katanin-type corrections [31] are included via the replacement $\dot{G}_{1,2}^\Lambda \to \frac{d}{d\Lambda} G_{1,2}^\Lambda$ instead. With certain approximations to the additional flow equation for the six-point vertex, two-loop contributions can also be added, see Appendix B for details. These equations are then solved numerically for the initial conditions $\Gamma_{1,2,3,4}^{\Lambda \to \infty} = V_{i_1 \alpha_1, i_2 \alpha_2, i_3 \alpha_3, i_4 \alpha_4}$, where the interaction $V$ is determined by the exchange couplings $J_{ij}$ in the present case.

Since the Majorana spin representation introduces no unphysical states, Eq. (2) can be used to study arbitrary temperatures. However, as discussed in Ref. [13] the artificial degeneracy of original spin states leads to a spurious Curie-type $1/T$ divergence of certain frequency components of vertices. The truncation of the flow equation causes these divergencies to affect the flow of frequency components related to spin correlations. Hence, unphysical results are obtained at $T = 0$ and small $\Lambda$. Below we demonstrate that this problem is significantly alleviated at finite (and not too low) temperatures, such that our approach can still be faithfully applied in temperature regimes where quantum and thermal fluctuations compete. The physical solution in the zero-cutoff limit $\Lambda = 0$ allows for the computation of temperature-dependent observables such as spin-spin correlations and -susceptibilities on the Matsubara axis,

$$\chi_{ij}(i\nu_n) = \int_0^\beta d\tau e^{-i\nu_n \tau} \left\langle S_i^z(\tau) S_j^z(0) \right\rangle,$$

$$\chi_q(i\nu_n) = \frac{1}{N} \sum_{i,j} e^{iq(r_i - r_j)} \chi_{ij}(i\nu_n). \tag{3}$$

The free energy per site $f$ can be found via Eq. (2a). Hence, temperature dependent thermodynamic quantities such as the energy per site, entropy and specific heat capacity are available via derivatives of $f(T)$. Alternatively, the energy can be determined from the expectation value of the Hamiltonian, which can be written in terms of equal time spin-spin correlators [17]

$$\langle H \rangle = \sum_{i<j} J_{ij} \left\langle S_i(0) S_j(0) \right\rangle \tag{4}$$

with $\left\langle S_i(0) S_j(0) \right\rangle = \sum_n \chi_{ij}(i\nu_n)$. Technical details of the numerical implementation are found in Appendix C.

# 3   Simple cubic lattice

We start by investigating the capability of the one-loop PMFRG in systems with well established magnetic long-range order. To this end, we study the Heisenberg model [Eq. (1)] on the simple cubic lattice and set the nearest-neighbor antiferromagnetic coupling to $J_1 = 1$. With no

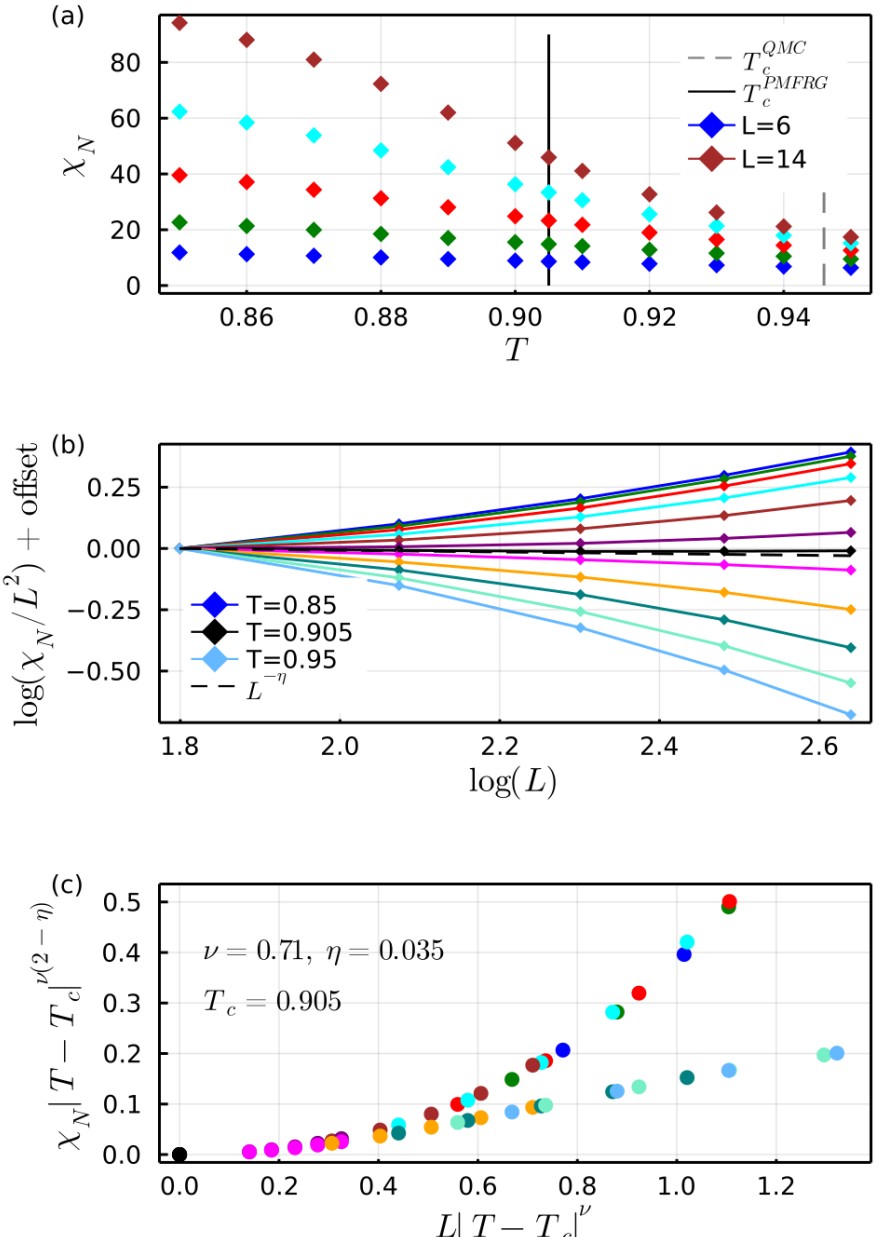

Figure 1: (a) Néel susceptibility from one-loop PMFRG in the antiferromagnetic nearest-neighbor Heisenberg model on the simple cubic lattice for temperatures around $T = 0.9$ and varying cutoff length $L = 6, 8, 10, 12, 14$. The number of positive Matsubara frequencies is $N_w = 32$. (b) Length-dependence of the susceptibility from (a); the critical temperature can be identified from a pure power-law behavior (no curvature in log-log plot). Adjacent curves have a temperature difference of $\Delta T = 0.01$, except of the black curve which has additionally been inserted for $T = 0.905$. (c) Scaling collapse for the data using the established critical exponents $\nu$ and $\eta$ from the classical 3D Heisenberg universality class.

further-neighbor couplings present, this model is unfrustrated and can be treated with the Quantum Monte-Carlo method (QMC). Sandvik [18] found magnetic Néel order with an ordering wavevector $\mathbf{q}_N = (\pi, \pi, \pi)$ below a critical temperature $T_c^{\text{QMC}} = 0.946(1)$. Finite-size scaling of the static Néel susceptibility $\chi_N$ computed for a cubic-box geometry with a linear size of up to $L_{\text{box}}^{\text{QMC}} \leq 16$ and periodic boundary conditions confirmed that the transition is in the classical 3D Heisenberg universality class with correlation length critical exponent $\nu = 0.71$ and anomalous dimension $\eta = 0.035$ known from Monte-Carlo simulations of numerically less demanding classical systems [32].

In the following, we benchmark the one-loop PMFRG against these well-controlled QMC results. In contrast to QMC, the PMFRG treats formally infinite (translational invariant) systems but introduces a cutoff-length $L$. Correlations between lattice sites with a distance larger than $L$ are neglected by setting the associated irreducible vertices $\Gamma$ to zero. Consequently, convergence in $L$ cannot be expected if the system features large or even divergent correlation length scales as, for example, close to a phase transition. While this effect has never been systematically studied in the context of PFFRG, here we turn it into an advantage and demonstrate that in the spin-FRG context $L$ can be used for finite-size scaling, just as the box size $L_{\text{box}}^{\text{QMC}}$ in the context of QMC.

Our PMFRG results for the static (end-of-flow) Néel-susceptibility $\chi_N$ around $T = 0.9$ and cutoff-lengths $L = 6, 8, 10, 12, 14$ are shown in Fig. 1(a). As expected, the missing convergence of $\chi_N$ with $L$ (except possibly at the largest $T$) indicates the presence of a correlation length larger than $L_{\text{max}} = 14$. Although this number seems modest we are treating about $4/3\pi L_{\text{max}}^3 \simeq 11494$ sites correlated to a reference site, almost three times the maximal number of sites considered in the QMC analysis of Ref. [18].

In Fig. 1(b), we determine the critical temperature from the expected behavior $\chi_N(T = T_c, L)/L^2 \propto L^{-\eta}$, which singles out the data trace for the critical temperature $T = T_c$ from the condition of vanishing curvature [1]. We find $T_c = 0.905(5)$, about 5% smaller than the QMC reference value $T_c^{\text{QMC}} = 0.946(1)$. In principle $\eta$ could be estimated independently from the slope of the $T_c$-data trace. In practice, this is difficult due to the limited system sizes in a quantum simulation and the numerically small value of $\eta = 0.035$, so that we are content with showing consistency between the measured and predicted slope (dashed line). In contrast, the value of the correlation length exponent $\nu$ is easier to confirm. In Fig. 1(c) we check the anticipated finite-size scaling behavior for temperatures $T$ in the vicinity of $T_c$ [18],

$$\chi_N(L, T) \propto |T - T_c|^{-\nu(2-\eta)} g_\pm \left( L|T - T_c|^\nu \right). \tag{5}$$

Using $T_c$ as obtained above, our PMFRG data nicely collapses into two branches of the scaling function $g_\pm$ for $T \gtrless T_c$.

We proceed by involving additional couplings between next-nearest and next-next-nearest neighbouring sites, $J_{2,3}$. Here, $J_2$ ($J_3$) is a coupling between sites separated along the face (body) diagonal of an elementary cube. The $J_1$-$J_3$ Heisenberg model is unfrustrated and can again be studied with QMC [27]. The PMFRG susceptibility for the case $J_3 = 0.4$ known to enter a Néel ordered phase, is shown in Fig. 2(a) and indicates a critical temperature $T_c = 1.875$, again about 5% different from the QMC value $T_c^{\text{QMC}} = 1.7675$.

Finally, we frustrate the system by a next-nearest neighbor coupling $J_2$. In the classical case, Monte-Carlo simulations [33] (with unit spin length) have found the phase diagram in Fig. 3, see blue symbols. Increasing $J_2$ from zero, the ordering temperature for Néel order decreases until it reaches $T_c \simeq 0.3$ at $J_2 = 0.25$ from where on a striped antiferromagnetic order with wave vector $(0, \pi, \pi)$ and equivalent types take over and the ordering temperature increases again. In the quantum case of $S = 1/2$ spins, where QMC suffers from the sign problem, the phase diagram

---

[1]For all scaling plots, we re-define $L = [3/(4\pi n)N]^{1/3}$ using the number $N$ of sites correlated to the reference site. The number of sites in unit volume is denoted by $n$, $n = 1$ for cubic- and $n = 16$ for the pyrochlore lattice. This smoothens edge-effects for small $L$ and yields better scaling plots.

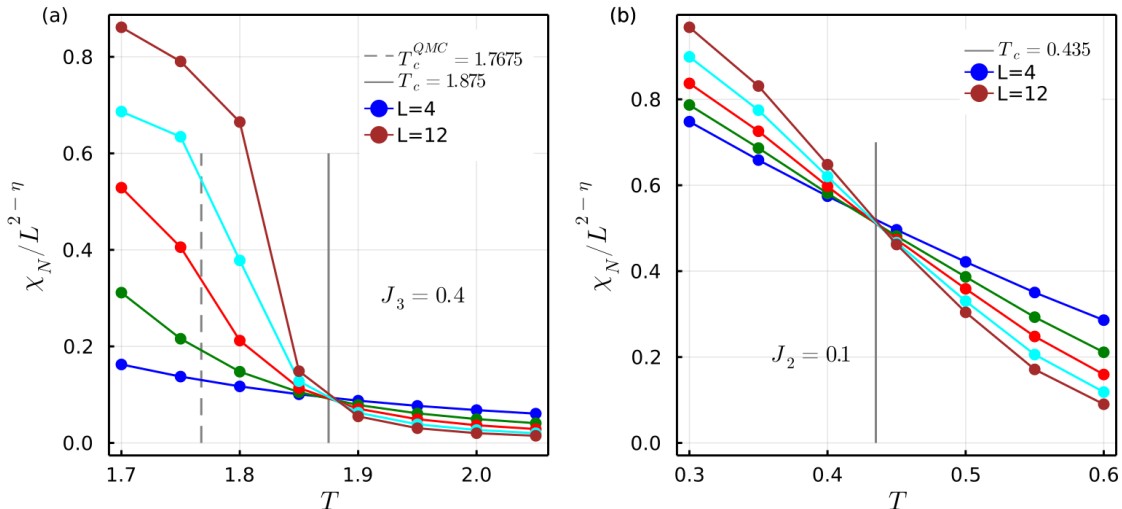

Figure 2: Scaling plot of the cubic lattice susceptibility similar to Fig. 1, but for the $J_1$-$J_3$ model with $J_3 = 0.4$ (a) and the frustrated $J_1$-$J_2$ model with $J_2 = 0.1$ (b). PMFRG estimates for the critical temperatures follow from the unique crossing points of the data traces.

has been studied with a variety of methods like spin-wave theory [22, 23], spherically symmetric Green's function approximation [24], differential operator technique [25], coupled cluster method [26] and the PFFRG [27]. Despite all these efforts, no consistent picture of the phase diagram has emerged. The qualitative question is if quantum fluctuations suppress the classical magnetic order around $J_2 = 0.25$ in favor of an intervening paramagnetic phase at $T = 0$. The PFFRG, for example, qualitatively reproduces the classical result with a finite break-down scale of the flow (see below) for all $J_2$, see brown curve in Fig. 3 [2]. The coupled cluster method, which infers ground state properties from extrapolation of observables found for finite-size clusters, shows some indication for a tiny paramagnetic phase around $J_2 \simeq 0.275$.

In this challenging setting, we now demonstrate the capability of the PMFRG to tackle frustrated systems by studying small $J_2 = 0.1$, for which, according to the scaling plot in Fig. 2(b), Néel order is detected below $T_c = 0.435$. This surprisingly small value of $T_c$ (at half the temperature estimated from the break-down scale of the PFFRG flow in Ref. [27]) might hint towards a larger paramagnetic region in the $J_2/J_1$-phase diagram of the model than previously thought. Indeed, repeating the calculation of $T_c$ for various $J_2$ between zero and 0.1, we extrapolate the observed linear-in-$J_2$ behavior of $T_c$ to find it vanishing around $J_{2,c} \simeq 0.19$ (red dots and red dashed line). Although this extrapolation has to be taken cautiously, it seems to indicate the onset of a quantum disordered phase significantly below the estimated value $J_{2,c} \simeq 0.275$ from the coupled cluster method of Ref. [26]. Interestingly, the scaling approach of the PMFRG susceptibility fails for larger $J_2$ where no line-crossings could be observed for the expected ordering wave vectors, despite the susceptibilities growing significantly with decreasing temperature (data not shown). We take this as an indication that the first-order transition observed in the classical case [33] is still governing the quantum model. We leave it to future work to analyze first-order transitions within the PMFRG and to further investigate the exciting possibility that the paramagnetic quantum phase in the $J_1$-$J_2$ cubic lattice antiferromagnet might be larger than previously thought.

To summarize this section, our results indicate that one-loop PMFRG is suitable to study finite-temperature magnetic phase transitions in 3D frustrated and unfrustrated Heisenberg systems. Although critical temperatures are a few percent off from QMC reference values, the susceptibility

---

[2]In PFFRG, a paramagnetic phase is found by adding a finite $J_3 > 0$, see Ref. [27]

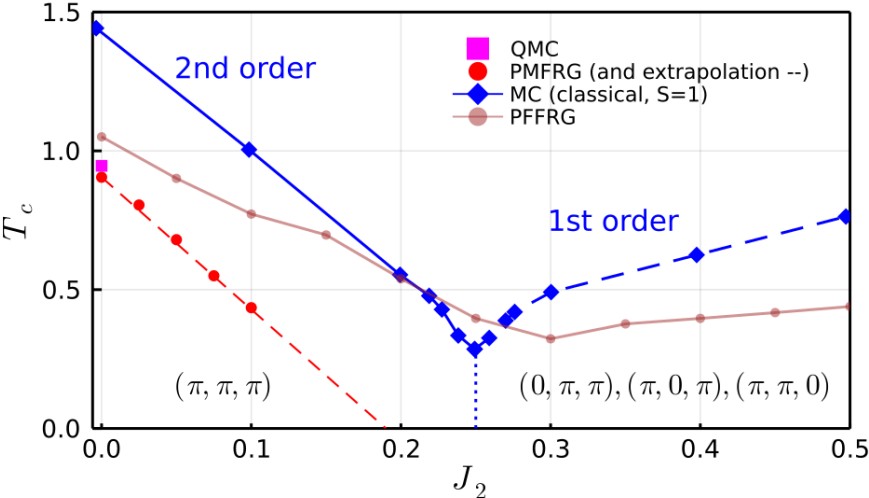

Figure 3: Finite temperature phase diagram of the simple cubic $J_1$-$J_2$ Heisenberg anti-ferromagnet. The data for the classical model with unit spin length is reproduced from Ref. [33] (blue), the transition to the Néel phase for $J_2 < 0.25$ is second order, while the striped phase for $J_2 > 0.25$ is reached via a first order transition. The PFFRG result reproduced from Ref. [27] is shown in brown. The one-loop PMFRG results (red dots) for ordering temperatures are only available for the second-order transition and at not too small temperatures; extrapolation to larger $J_2$ (red dashed line) yields $J_{2,c} \simeq 0.19$ at $T = 0$.

data shows the expected scaling behavior at second-order phase transitions, a strong indication for the beyond-mean-field nature of the PMFRG. In particular, there is no breakdown of the flow or any divergence in the susceptibility at any temperature treated. This is expected in the exact (or at least beyond-mean-field/RPA) treatment of an effectively finite-sized system which should not show any spontaneous symmetry breaking. The observed scaling behavior provides a significantly more accurate and rigorous approach to detect magnetic phase transitions than previous PFFRG works where kinks in the renormalization group flow have been used as a signature for ordering. Furthermore, estimates for critical temperatures within PFFRG were previously based on the approximate (i.e. mean-field-like) relation $T = \pi\Lambda/2$ between temperature and the cutoff scale which may introduce errors, particularly in the presence of strong quantum fluctuations. We thus firmly believe that it is advantageous to obtain finite ordering temperatures for frustrated models from PMFRG instead.

# 4 Pyrochlore lattice

While the previous section demonstrated the PMFRG's applicability to systems ordering magnetically, strongly frustrated and magnetically disordered models are also treatable. A prominent example of a geometrically frustrated lattice is the pyrochlore network [20], defined by a four-site basis arranged within an fcc lattice. Here, each site is placed at the vertex of an arrangement of corner-sharing tetrahedra where the edges are given by nearest-neighbor bonds [34]. The classical nearest-neighbor antiferromagnetic Heisenberg model features an extensive ground state degeneracy as the lowest energy can be achieved by any state fulfilling the constraint of a vanishing magnetization within each individual tetrahedron, often referred to as a *spin-ice rule* [35–37]. The quantum versions of models with such a degeneracy are often believed to evade magnetic long-

range ordering at low temperatures and, as such, are promising candidates as hosts for quantum spin liquids. Recent studies confirm the non-magnetic ground state of the nearest neighbor spin-1/2 pyrochlore antiferromagnet but suggest a spontaneous breaking of $C_3$ and inversion symmetry [38–40] possibly indicating a valence-bond solid. Yet, the predictions of magnetic monopole and emergent photon excitations resulting from an underlying $U(1)$ gauge structure remain a fascinating research perspective for related models [41]. Arising from the local nature of the ground state constraint, an interesting feature is the observation of non-analytical points in the classical spin structure factor, so-called "pinch-points" (also referred to as "bow-ties"), at $T = 0$ within the $hhl$-plane [42–44].

Being well-suited to treat quantum systems at finite temperatures, we now investigate the performance of the PMFRG in the case of the nearest-neighbor quantum spin-1/2 pyrochlore antiferromagnet. In order to verify the quantitative reliability of our results, we start comparing the static component of our spin susceptibility $\chi \equiv \chi(q = 0)$ against DMRG [40] and diagrammatic Monte-Carlo [45] as well as the Padé approximant of the high-temperature series expansion (HTSE) in Fig. 4 [46, 47]. On the one-loop level our results differ from other methods by $\sim 10\%$ at $T \sim J_1$ with further increasing differences for lower temperatures, indicating a smaller accuracy than the one-loop results in Sec. 3. However, under the additional inclusion of two-loop ($\ell = 2$) contributions our results are found to be in perfect agreement with all other methods, remaining consistent with DMC even at temperatures as low as $T \simeq 0.2$.

Figure 5 shows the energy per site $\epsilon$ and the specific heat capacity $c = \frac{d\epsilon}{dT}$ as functions of the temperature. It can be seen that the energy computed from the PMFRG susceptibility via Eq. (4) is generally consistent with the one derived from the PMFRG free-energy and HTSE, although acquiring an unphysical negative slope (i.e. negative heat capacity) around $T \lesssim 0.3$. This is likely a first indicator of the aforementioned low-temperature divergence in the PMFRG flow discussed above and in Ref. [13]. The energies obtained via the free energy, by contrast, retain a positive slope down to lower temperatures but will ultimately behave similar due to the free energy's indirect coupling to the four-point vertex. Despite this observation, we stress that the energy is not to be understood as a measure of accuracy in the variational sense and as such is not bounded from below by the true energy. While a temperature below $T \simeq 0.2J_1$ is currently not accessible, the finite temperature energy compares well with a recent many-variable Monte Carlo (mVMC) study at $T = 0$ (dashed black line).

**Finite-width pinch-points**

The spin susceptibility of the pyrochlore features bow-tie patterns in the $hhl$-plane, connected to the existence of the classical ground state ice rule [44]. In Fig. 6 we show the static susceptibility [Eq. (3)] from two-loop PMFRG at $T = 0.2$ in the $hhl$-plane, which features a pronounced peak structure around $q = (0, 0, 4\pi)$ (and symmetry-related points) where one would classically expect the pinch points. In the classical case, the width of these peaks along the [$00l$]-direction is known to vanish analytically in the $T \to 0$ limit whereas thermal fluctuations at $T > 0$ lift the non-analyticity of the pinch points. The associated finite width $\sigma \sim \sqrt{T}$ of the broadened peaks is a measure for how much the ice rule is violated at finite temperatures [21, 48, 49].

In a quantum system, thermal- and quantum fluctuations compete. Using the PMFRG, we measure the full-width at half maximum (FWHM) of the peak along the [$00l$]-direction, see Fig. 6(b). Although at low temperatures, we observe a straight line in a plot over $\sqrt{T}$, an extrapolation to $T = 0$ results in a finite width at $T = 0$ where two-loop PMFRG predicts a slightly smaller value than one-loop. It can be concluded that while the qualitative applicability of the classical ice rule remains visible in the overall structure of the susceptibility, a full $\sqrt{T}$-law without a constant offset is only correct for the classical model. Quantum effects not only broaden the peak at $T = 0$ [50], but remain strong enough at finite temperatures to increase deviations from the classical ice rule ground state.

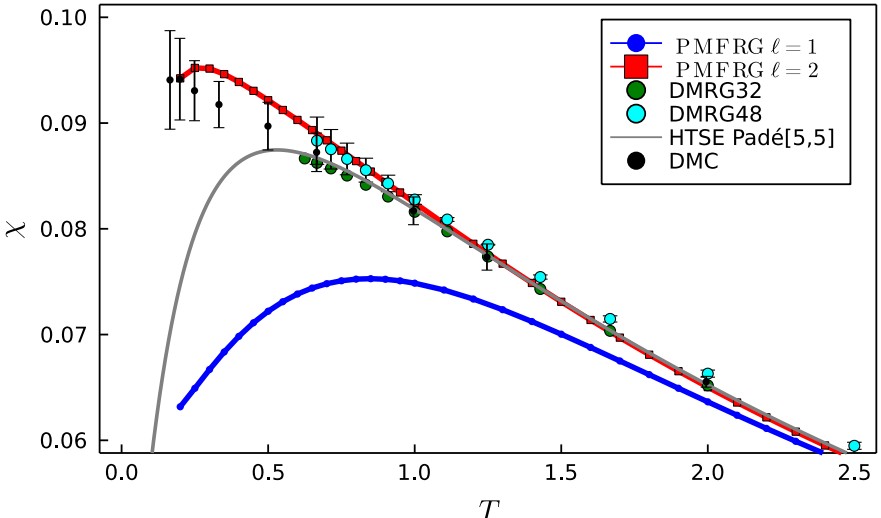

Figure 4: Uniform ($q = 0$) susceptibility for the Pyrochlore antiferromagnet from PMFRG as a function of temperature in comparison with diagrammatic Monte Carlo (DMC) [45], density-matrix renormalization group [40] (DMRG, cluster sizes 32 and 48) and the Padé approximant of the high temperature series expansion [46].

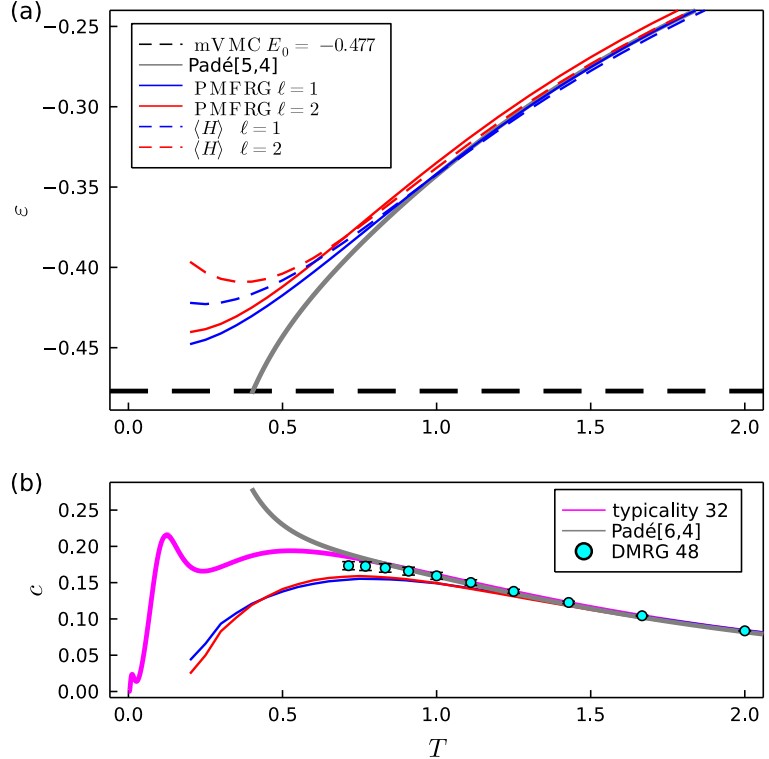

Figure 5: Energy per site (a) and specific heat capacity (b) as a function of temperature for one-loop ($\ell = 1$) in blue and two-loop ($\ell = 2$) in red. An estimate of the energy per site within PMFRG is accessible either from a derivative of the free energy (solid), Eq. (2a), or through the expectation value of $H$ in terms of equal time spin-correlators (dashed). Additionally shown is the ground state energy estimate from mVMC [39] and the specific heat capacity from DMRG and canonical typicality on a 48- and 32-site cluster, respectively [40].

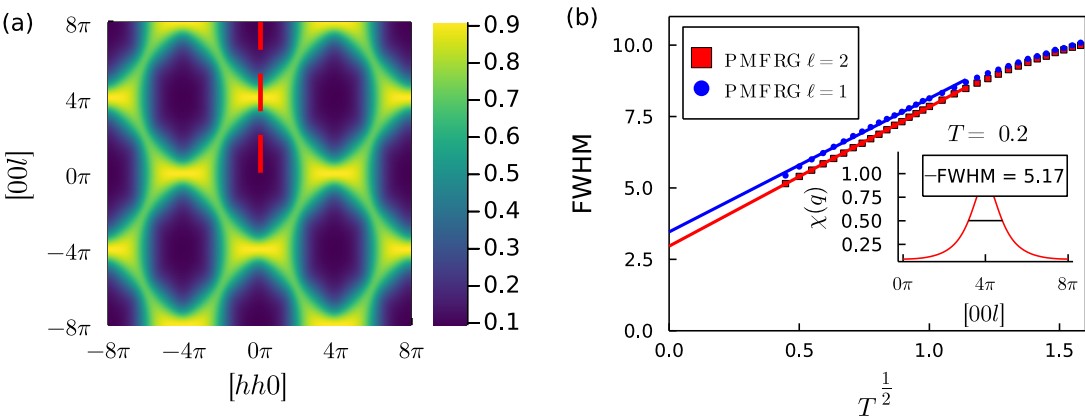

Figure 6: (a) Two-loop static susceptibility of the antiferromagnetic Heisenberg model on the pyrochlore lattice in the $[hhl]$-plane ($q_x = q_y = h$) at $T = 0.2$ and (b) full-width at half maximum along of the pinch-point as a function of temperature. The inset shows the cut along the $[00l]$ line of the susceptibility from (a).

## 5   Effect of two-loop contributions

Our results in Sec. 3 demonstrate that one-loop PMFRG allows one to accurately determine critical temperatures and scaling behavior for second order magnetic phase transitions in 3D quantum magnets. On the other hand, in strongly frustrated systems that remain magnetically disordered at low temperatures such as the pyrochlore model investigated in the last section, one-loop results are less accurate but two-loop corrections yield substantial improvements. What remains to be discussed is how two-loop PMFRG performs when applied to magnetically ordered systems.

To demonstrate the two-loop flow behavior in this case, we specifically consider the ferromagnetic ($J_1 = -1$) nearest neighbor pyrochlore Heisenberg model but emphasize that the results below are typical for magnetically ordered systems. While as usual the susceptibility flows smoothly as a function of the cutoff $\Lambda$ (see Fig. 7), the one-loop susceptibility scales strongly with system size yielding a critical temperature $T_c \simeq 0.685$ in good agreement to QMC ($T_c^{\mathrm{QMC}} = 0.7182$ [49]), see the crossing lines in Fig. 8. However, for $\ell = 2$, no such scaling and, hence, no magnetic order is found. The large quantitative difference between one-loop and two-loop in the magnetically ordered case suggests the necessity for higher loop order corrections, which we leave for future work.

Initially, it may appear surprising that the detection of magnetic order is problematic at second loop order. However, a similar observation has been made in a recent multiloop PFFRG study [16], where magnetic ordering tendencies in the flow are found to be strongly suppressed at $\ell = 2$ but recovered at $\ell = 3$.

A deeper understanding of this behavior can be obtained by inspecting the diagrammatic contributions in different loop orders. First recall that the four-point vertex flow is generated by different coupling channels with distinct physical meanings. Particularly, the random-phase approximation (RPA) terms enable the formation of magnetic long-range order, while all other channels (here, for simplicity referred to as "ladder channels") induce quantum fluctuations. In multi-loop schemes these channels are inserted into each other, leading to a nested diagram structure, see Fig. 7 for an example. The nesting is subject to the rule that a contribution from a particular channel cannot be inserted into the same channel again, as this would yield an overcounting of diagrams.

With this multiloop construction in mind, the RPA diagrams which in magnetically ordered systems dominate the one-loop flow are dressed by ladder diagrams in two-loop. This strongly

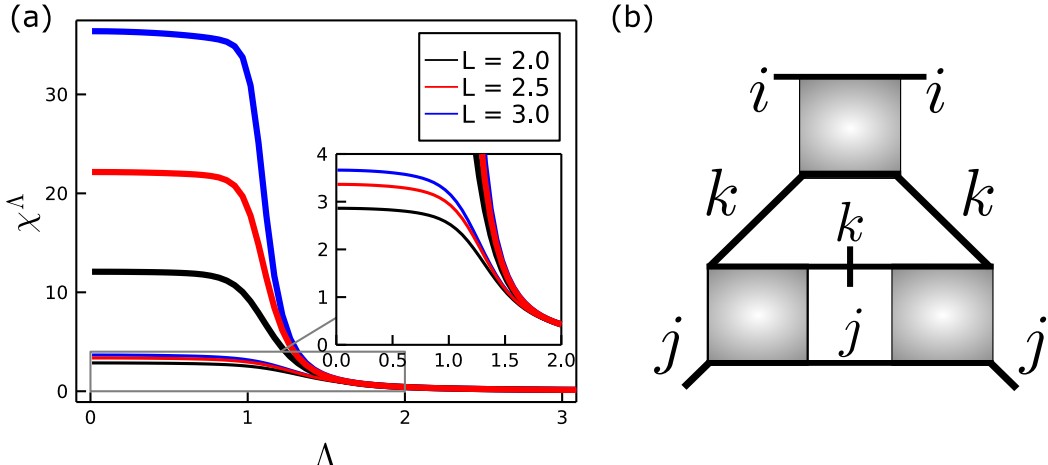

Figure 7: (a) Ferromagnetic Heisenberg model on the pyrochlore lattice for $T = 0.5$, well below the critical temperature $T_c = 0.685$ observed in Fig. 8: Flow of the uniform susceptibility $\chi^\Lambda$ obtained in the one-loop (thick) and two-loop (thin) PMFRG as a function of the cutoff $\Lambda$ for different maximum vertex lengths $L$. (b) Two-loop contribution to the right hand side of the flow equation for $\Gamma$ where a ladder diagram (with external site indices $k$, $j$) is inserted into the RPA channel (with external site indices $i$, $j$).

suppresses magnetic order and explains our observation in Fig. 7. In turn, the third loop order nesting can again be performed with RPA diagrams which would strengthen ordering effects. Overall, one may, hence, expect an even-odd-effect of magnetic ordering tendencies in loop order. We believe that this type of behavior is characteristic for systems where one coupling channel (here, the RPA channel) dominates the physical behavior. The more systematic improvement upon increasing $\ell$ observed for the magnetically disordered antiferromagnetic pyrochlore Heisenberg model can then be interpreted as a consequence of the fact that in this case *all* channels contribute more equally. In both situations, an increase of loop order should eventually yield more accurate results but possibly not in a monotonous way. The case $\ell \geq 3$, however, is beyond the scope of the present work and will be left for future studies.

# 6 Conclusion

In this work, we applied the PMFRG to unfrustrated and frustrated 3D Heisenberg quantum spin systems and demonstrated, based on a variety of different physical quantities, an astonishing quantitative accuracy of this technique. Rigorous benchmark tests were performed by comparing our results for the unfrustrated simple cubic lattice antiferromagnet with error controlled QMC data. We found that the PMFRG can keep up with QMC's performance for these systems, producing errors of about 5% for the critical ordering temperatures and showing overall consistency for the critical exponents $\nu$ and $\eta$. A special methodological feature of our scaling strategy is its reference to a cutoff length $L$ for spin correlations in an infinite system but not to the size of a box containing the simulated spins.

We have also investigated frustrated systems such as the nearest neighbor pyrochlore antiferromagnet and the $J_1$-$J_2$ simple cubic lattice antiferromagnet. While possibilities for quantitative comparisons with other methods become rarer we found promising indications that the PMFRG's performance persists for magnetically disordered frustrated systems, at least when including two-loop corrections. Particularly, we demonstrated this for the two-loop static and homogeneous

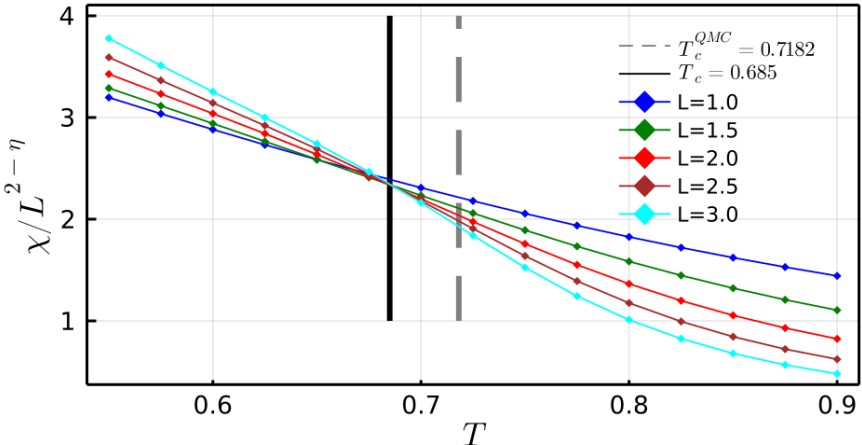

Figure 8: PMFRG ($\ell = 1$) results for the ferromagnetic Heisenberg model on the pyrochlore lattice indicating a phase transition at $T_c \simeq 0.685$ in good agreement with the QMC value $T_c^{\mathrm{QMC}} = 0.7182$ from Ref. [49].

susceptibility of the nearest neighbor pyrochlore antiferromagnet where our data is within the error bars of DMC over nearly the entire simulated temperature range. Energies per site, which are no standard outputs of functional renormalization group approaches and are, therefore, rarely studied, likewise, show good accuracies and seem consistent with the ground state energies from other methods. We also made first attempts to characterize ground state phases, e.g., an extrapolation of the width of pinch-points in the nearest neighbor pyrochlore antiferromagnet clearly shows a residual broadening in the limit $T \to 0$ as has previously been found with various other approaches [16, 50, 51]. Furthermore, ordering temperatures drop surprisingly fast in the simple cubic lattice antiferromagnet upon adding second neighbor interactions $J_2$, possibly indicating a non-magnetic ground state regime.

Our work opens up a variety of possibilities for future applications of the PMFRG. Having already implemented a two-loop scheme, the natural next step is the inclusion of higher loop orders with $\ell \geq 3$. We expect that this eventually increases the accuracy of our approach further, especially towards lower temperatures. However, our results for magnetically ordered systems where one-loop scaling behavior is erased in a two-loop extension, implies a non-trivial behavior in loop order $\ell$ such that convergence in $\ell$ might turn out to be technically challenging. Note that similar observations have already been made with PFFRG [16, 17]. We argued that the accuracy in loop order for magnetically ordered systems might be subject to an even-odd effect while magnetically disordered systems are expected to be more well-behaved as a function of $\ell$.

An important advantage of the PMFRG over the PFFRG is that it allows the detection of second order phase transitions in a completely unambiguous and rigorous way via finite-size scaling. We, therefore, believe that the investigation of critical behaviors within PMFRG represents a promising future research direction. Interestingly, the absence of finite-size scaling in the $J_1$-$J_2$ simple cubic Heisenberg model at $J_2 > 0.25$ within one-loop PMFRG is consistent with a first order transition in the corresponding classical model. This may indicate that already in one-loop our method is capable of distinguishing between first and second order phase transitions. However, such questions or the precise detection of first order transitions certainly require further investigation. Eventually, at zero temperature, the detection of quantum criticality in two dimensions remains an open problem, particularly for Heisenberg models where the Mermin-Wagner theorem forbids any magnetic order at finite temperatures.

Concluding this work with a broader perspective, we emphasize that the PMFRG inherits the

same methodological flexibility that already characterizes the PFFRG. This means that the method is amenable to arbitrary lattice geometries and two-body spin interactions. The implementation of spin-anisotropic couplings also requires only moderate adjustments. In this situation, applications to models for real magnetic materials beyond the ideal systems studied here are well within reach.

# 7    Acknowledgements

We thank Dominik Kiese, Vincent Noculak, Lode Pollet and Marc Ritter for useful discussion. We acknowledge Imre Hagymási for his kind supply of the DMRG data and Yasir Iqbal for sharing PFFRG data reproduced from Ref. [27] in Fig. 3. NN acknowledges funding from the German Research Foundation within the TRR 183 (project A04). BS acknowledges support from the German National Academy of Sciences Leopoldina through Grant Number LPDR 2021-01, from a MCQST-START fellowship and from the Munich Quantum Valley, which is supported by the Bavarian state government with funds from the Hightech Agenda Bayern Plus. Numerical computations have been performed on the CURTA cluster at FU Berlin [52].

# A    Inclusion of the RPA

In order to investigate the PMFRG's behaviour regarding a magnetic phase transition, we consider the contributions of the RPA channel in the (one-loop) PMFRG flow equations. We can do this mostly in analogy to Ref. [10], except that we now explicitly consider finite temperatures. In the RPA approximation for PMFRG, we restrict ourselves to diagrams with internal Majorana bubbles, i.e. site summations. As a result, the flow equations for the three types of vertices as presented in Ref. [13] decouple from each other. As seen in Eq. (26) the only vertex which is nonzero initially is the spin vertex $\Gamma_c = \Gamma_{xyxy}$,

$$\frac{d}{d\Lambda}\Gamma^\Lambda_{c\,ij}(s,t,u) = T\sum_\omega \dot{g}^\Lambda(\omega)g^\Lambda(\omega+s)\sum_k \Big[\Gamma^\Lambda_{c\,ki}(s,\omega+\omega_1,\omega+\omega_2)\,\Gamma^\Lambda_{c\,kj}(s,\omega-\omega_3,\omega-\omega_4)$$
$$+ (\omega_1 \leftrightarrow \omega_2, \omega_3 \leftrightarrow \omega_4)\Big]. \tag{6}$$

Since the vertices of type $\Gamma_a$ and $\Gamma_b$ are vanishing, it follows that the self energy must be zero as well and thus

$$g^\Lambda(i\omega_n) = \frac{\omega_n}{\omega_n^2 + \Lambda^2},$$
$$\dot{g}^\Lambda(i\omega_n) = -\frac{2\Lambda}{\omega_n}g^2(i\omega_n). \tag{7}$$

Using that $\Gamma^{\Lambda\to\infty}_{c\,ij} = -J_{ij}$ does not depend on any frequencies, we note that no dependence on $t$ and $u$ is generated from Eq. (6). The dominant contribution is the static component $\Gamma^\Lambda_{c\,ij}(s=0) \equiv \Gamma^\Lambda_{c\,ij}$ for which

$$\frac{d}{d\Lambda}\Gamma^\Lambda_{c\,ij} = -4\Lambda\sum_k \Gamma^\Lambda_{c\,ki}\Gamma^\Lambda_{c\,kj}T\sum_\omega \frac{(g^\Lambda(\omega))^3}{\omega},$$
$$\frac{d}{d\Lambda}\Gamma^\Lambda_c(\boldsymbol{k}) = -4\Lambda\Gamma^\Lambda_c(\boldsymbol{k})^2 T\sum_n \frac{\omega_n^2}{(\omega_n^2+\Lambda^2)^3},$$
$$\tag{8}$$

where in the second step a Fourier transform to momentum space has been performed. The Matsubara sum may be evaluated exactly using the poles $z_p \equiv i\omega_n = \pm\Lambda$ to obtain

$$T \sum_n \frac{\omega_n^2}{(\omega_n^2 + \Lambda^2)^3} = \sum_{z_p = \pm\Lambda} \text{Res}\left(\frac{z^2}{(z^2 - \Lambda^2)^3} n_F(z)\right)\Bigg|_{z=z_p}$$
$$= \frac{\text{sech}^2\left(\frac{\beta\Lambda}{2}\right)\left(\sinh(\beta\Lambda) + \beta\Lambda\left(\beta\Lambda\tanh\left(\frac{\beta\Lambda}{2}\right) - 1\right)\right)}{32\Lambda^3}. \tag{9}$$

Inserting this result into Eq. (8), the differential equation with $\Gamma_c^{\Lambda\to\infty}(\boldsymbol{k}) = -J(\boldsymbol{k})$ has the exact solution

$$\Gamma_c^\Lambda(\boldsymbol{k}) = -\frac{8J(\boldsymbol{k})\Lambda}{2J(\boldsymbol{k})\tanh\left(\frac{\beta\Lambda}{2}\right) + \beta J(\boldsymbol{k})\Lambda\text{sech}^2\left(\frac{\beta\Lambda}{2}\right) + 8\Lambda},$$
$$\left[\Gamma^{\Lambda=0}(\boldsymbol{k})\right]^{-1} = -\frac{1}{4T} - \frac{1}{J(\boldsymbol{k})}, \tag{10}$$

in the simplified case of a single site per unit cell.

Below a critical temperature $T_c^{\text{RPA}} = \frac{1}{4}J(\boldsymbol{k})$, the RPA-vertex from Eq. (10) diverges before the end of the flow at $\Lambda = 0$ is reached. This result exactly equals the one derived in Ref. [10], except here, no identification of $\Lambda_c$ with $T_c$ is necessary as Eq. (10) has been derived directly for arbitrary temperatures. Figure 9 shows the flow of the RPA vertex in a nearest-neighbor cubic lattice where $T_c^{\text{RPA}} = 1.5$.

Interestingly, our full PMFRG solution is in stark contrast to bare RPA: While we could show here that the RPA's individually diverging contributions are contained in the PMFRG, no divergence at finite $\Lambda$ is observed, in favor of a finite and smoothly flowing susceptibility as shown in Fig. 7. This beyond mean-field nature of the PMFRG, a result of the additional contributions from other channels, is quite surprising: In the closely related PFFRG formalism, a divergence of the RPA channel is often observed and, in particular, serves as the main indicator for the onset of magnetic order. In Section 3, we demonstrated that the absence of such an RPA-like divergence is extremely beneficial: The finite susceptibility which becomes physical at $\Lambda = 0$ can be used in combination with a finite-size scaling analysis to obtain a more accurate estimate of critical temperatures.

## B  Two-loop contributions within PMFRG

As detailed in previous works [53, 54], the one-loop FRG truncation can be extended by the inclusion of two-loop corrections using approximations based on the flow equation of the six-point vertex.

We start from the general form of the FRG flow equations, as found in Eq. (7.71) of Ref. [5] and expand the summations, neglecting all contributions from vertices with an odd number of legs as well as the eight-point vertex. For Majorana systems, the exchange statistics implies $Z = -1$ so that

$$\frac{d}{d\Lambda}\Gamma_{1,2,3,4,5,6}^{6\,\Lambda} = \frac{1}{2}\text{Tr}\Big[S_{1,2,3,4|5,6}\dot{G}^\Lambda\Gamma_{5,6}^{4,\,\Lambda}G^\Lambda\Gamma_{1,2,3,4}^{6,\,\Lambda} \tag{a}$$
$$+ S_{1,2|3,4,5,6}\dot{G}^\Lambda\Gamma_{3,4,5,6}^{6,\,\Lambda}G^\Lambda\Gamma_{1,2}^{4,\,\Lambda} \tag{b}$$
$$+ S_{1,2|3,4|5,6}\dot{G}^\Lambda\Gamma_{5,6}^{4,\,\Lambda}G^\Lambda\Gamma_{3,4}^{4,\,\Lambda}G^\Lambda\Gamma_{1,2}^{4,\,\Lambda}\Big] \tag{c}$$
$$+ \mathcal{O}(V_{\text{int}}^4). \tag{11}$$

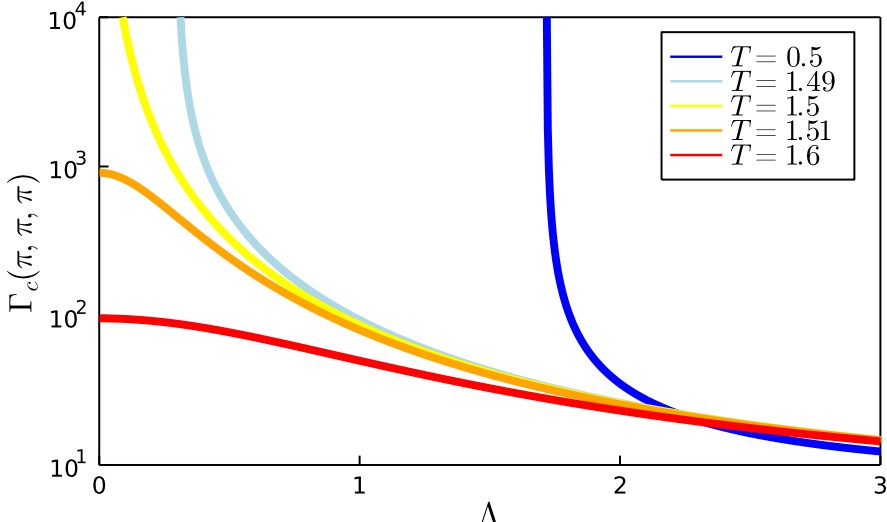

Figure 9: RPA solutions as a function of the cutoff $\Lambda$ from Eq. (10) for the nearest-neighbor cubic lattice for different temperatures. The solution for the critical temperature $T_c^{\mathrm{RPA}} = \frac{6}{4} = 1.5$ (green) diverges exactly at $\Lambda = 0$, while at lower temperatures, the divergence is shifted to finite cutoffs.

Bold quantities are matrices defined as $\left[\mathbf{\Gamma}^{6,\,\Lambda}_{1,2,3,4}\right]_{5,6} = \Gamma^{6,\,\Lambda}_{5,6,1,2,3,4}$.

This expression further contains the symmetrization operator $S$ which ensures that the derivative of the six-point vertex is fully antisymmetric. Formally, it can be written as a sum over all permutations of indices with the appropriate sign together with a prefactor to prevent overcounting of already antisymmetric terms. For instance, the symmetrization $S_{1,2,3,4|5,6}$ in term (a) of Eq. (11) contains a summation over all permutations of the numbers 1 to 6 as well as a prefactor $\frac{1}{4!2!}$ since the expression is already antisymmetric in the first four and the last two indices. If we define the *outer* derivative $\partial_\Lambda$ which only acts on the explicit $\Lambda$-dependence of two-point Green's functions(treating $\Sigma_\Lambda$ as a constant), we may write this as

$$
\frac{d}{d\Lambda}\Gamma^{6\,\Lambda}_{1,2,3,4,5,6} = \frac{1}{2}\sum_{1',\ldots,4'}\left[\partial_\Lambda\left(G^\Lambda_{1'2'}G^\Lambda_{3'4'}\right)S_{1,2|3,4,5,6}\Gamma^{6,\,\Lambda}_{2',3',3,4,5,6}\Gamma^{4,\,\Lambda}_{4',1',1,2}\right]
$$

$$
+\frac{1}{6}\sum_{1',\ldots,6'}\left[\partial_\Lambda\left(G^\Lambda_{1'2'}G^\Lambda_{3'4'}G^\Lambda_{5'6'}\right)S_{1,2|3,4|5,6}\Gamma^{4,\,\Lambda}_{2',3',1,2}\Gamma^{4,\,\Lambda}_{4',5',3,4}\Gamma^{4,\,\Lambda}_{6',1',5,6}\right] + \mathcal{O}(V^4_{\mathrm{int}}) \quad (12)
$$

The defining step of the two-loop scheme is to promote the partial derivative to a full one which, in particular, also acts on vertex functions. The error generated by this step is of order $\mathcal{O}(V^4)$ in the interaction and thus no larger than the error already present [53, 54]. The resulting equation can be integrated as a function of $\Lambda$ and leads to a self-consistent equation for $\Gamma^6$ for which in first iteration, we get

$$
\Gamma^{6\,\Lambda}_{1,2,3,4,5,6} = \frac{1}{12}\sum_{1',\ldots,4'}\sum_{\beta_1,\ldots,\beta_6}\left[G^\Lambda_{1'2'}G^\Lambda_{3'4'}S_{1,2|3,4,5,6}\Gamma^{4,\,\Lambda}_{4',1',1,2}\right.
$$

$$
\times\left(G^\Lambda_{\beta_1\beta_2}G^\Lambda_{\beta_3\beta_4}G^\Lambda_{\beta_5\beta_6}S_{1',2'|3,4|5,6}\Gamma^{4,\,\Lambda}_{\beta_2,\beta_3,2',3'}\Gamma^{4,\,\Lambda}_{\beta_4,\beta_5,3,4}\Gamma^{4,\,\Lambda}_{\beta_6,\beta_1,5,6}\right)\Big]
$$

$$
+\frac{1}{6}\sum_{1',\ldots,6'}\left[\left(G^\Lambda_{1'2'}G^\Lambda_{3'4'}G^\Lambda_{5'6'}\right)S_{1,2|3,4|5,6}\Gamma^{4,\,\Lambda}_{2',3',1,2}\Gamma^{4,\,\Lambda}_{4',5',3,4}\Gamma^{4,\,\Lambda}_{6',1',5,6}\right] + \mathcal{O}(V^4_{\mathrm{int}}). \quad (13)
$$

Figure 10 shows the diagrammatic form of this equation. While the first term is of fourth order

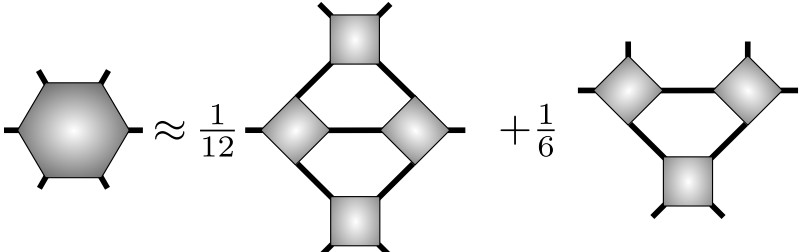

Figure 10: Two-loop approximation for the six-point vertex.

in the interaction and will not be considered explicitly, we note that some of its contributions are precisely those generated by the Katanin substitution as detailed in Ref. [53].

In the same way, some of the derived two-loop contributions are equivalent to Katanin corrections of the one-loop flow equations. Naturally, the next step will be to identify these terms and omit them to prevent overcounting. Doing so requires explicitly evaluating all permutations generated by the symmetrization operator $S$. Initially, now using the shorthand notation $\Gamma^4 \to \Gamma$, we thus have

$$
\frac{d}{d\Lambda}\Gamma^\Lambda_{1,2,3,4} \equiv \dot{\Gamma}^{\Lambda\,1L}_{1,2,3,4} + \dot{\Gamma}^{\Lambda\,2L}_{1,2,3,4}
$$
$$
\dot{\Gamma}^{\Lambda\,2L}_{1,2,3,4} = -\frac{1}{12}\sum_{1',2'}\dot{G}^\Lambda_{1',2'}\sum_{\beta_1,\dots,\beta_6}\left[\left(G^\Lambda_{\beta_1\beta_2}G^\Lambda_{\beta_3\beta_4}G^\Lambda_{\beta_5\beta_6}\right)S_{1',2'|1,2|3,4}\Gamma^\Lambda_{\beta_2,\beta_3,1',2'}\Gamma^\Lambda_{\beta_4,\beta_5,1,2}\Gamma^\Lambda_{\beta_6,\beta_1,3,4}\right].
$$
(14)

Here, $\dot{\Gamma}^{\Lambda\,1L}_{1,2,3,4}$ refers to the three one-loop terms in Eq. (2c), which do not originate from the six-point vertex.

Expanding the symmetrization naïvely generates $6! = 720$ permutations, however many of these are equivalent. Most importantly, all trivial permutations that exchange two indices on the same vertex are divided out by definition of $S$. This means we only need to consider $\frac{720}{2!\cdot2!\cdot2!} = 90$ terms. Since we do not want to include terms which are given by the Katanin correction to the one-loop procedure, we will then neglect all diagrams in which a single vertex is contracted by the single-scale propagator, i.e. those where $1'$ and $2'$ appear on the same vertex. Hence, only 72 diagrams remain, 24 for each the $s, t$ and $u$ channel.

It is helpful to note that $t$ and $u$ channels are given by re-labeling external indices of the $s$-channel, i.e. the first of the terms in Eq. (2c). Thus, we only need to consider the $s$-channel, which is defined by a pairing of either the indices 1 and 2 on one of the vertices or 3 and 4. Using the freedom to relabel internal site indices in the summation, only two distinct diagrams remain, one where 1 and 2 appear together on a vertex and the other two appear separately on the other two vertices and vice versa. In close analogy to the previous one-loop notation, we may then define

$$
\dot{\Gamma}^{\Lambda\,2L}_{1,2,3,4} = Y^\Lambda_{1,2|3,4} - Y^\Lambda_{1,3|2,4} + Y^\Lambda_{1,4|2,3}
$$
(15)

$$
Y^\Lambda_{1,2|3,4} = -\sum_{1',\dots,4'}G^\Lambda_{1',2'}G^\Lambda_{3',4'}\left(\Gamma^\Lambda_{1,2,4',2'}X^\Lambda_{3,3'|4,1'} + \Gamma^\Lambda_{1',3',3,4}X^\Lambda_{2',1|4',2}\right),
$$
(16)

where $Y^\Lambda_{1,2|3,4}$ defines the s-channel of the two-loop bubble function and is antisymmetric under permutations of the first and last two indices as visible from Fig. 11. Since Eq. (16) takes an analogous expression as the one-loop equations, using pre-computed one-loop bubble functions, computing the two-loop contributions amounts the same numerical complexity as the one-loop terms and thus approximately doubles the numerical effort.

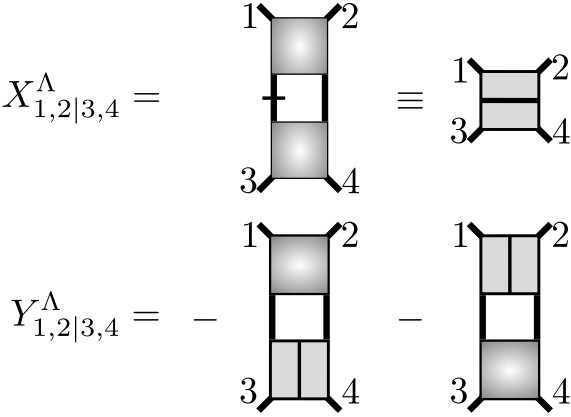

Figure 11: One-loop and two-loop bubble functions from Eqs. (2d) and (16).

## B.1   Parametrization

As usual, an efficient implementation requires the explicit parametrization of vertices in analogy to Ref. [13]. This parametrization is equivalent for both the one-loop ($X$) and the two-loop bubble-functions $Y$ so that for brevity we shall only write the results for $X$ explicitly. It is evident from their definitions that the bilocal property of vertices carries over to $X$ and $Y$ due to the local nature of propagators. In the case of vertices, it is possible to re-arrange indices such that they are always of the form $\Gamma^\Lambda_{iijj}$, however, for $X$ and $Y$ only the first and last two indices may be interchanged and hence we need to distinguish two distinct types of bubble-functions upon real-space parametrization

$$X^\Lambda_{ij} \equiv X^\Lambda_{ii|jj}$$
$$\tilde{X}^\Lambda_{i\neq j} \equiv X^\Lambda_{ij|ij}, \quad \tilde{X}^\Lambda_{ii} = X^\Lambda_{ii}. \tag{17}$$

Physically, $X^\Lambda_{ij}$ corresponds to an RPA-type contribution in which a summation over all sites occurs. This can be seen from Fig. 11, where after external site indices are inserted, the propagators carry an internal site index $k$ which may differ from both $i$ and $j$ in contrast to $\tilde{X}_{ij}$. Furthermore, energy conservation implies $X(\omega_1, \omega_2|\omega_3, \omega_4) \propto \delta_{\omega_1+\omega_2+\omega_3+\omega_4,0}$ and equally for $Y$ which allows the usual definition via only three exchange frequencies $s, t, u$. Subsequently, summations over flavors may be computed explicitly by making use of the global $SO(3)$ symmetry to distinguish three $X$-types $X_a, X_b, X_c$ and four $\tilde{X}$-type vertices $\tilde{X}_a, \tilde{X}_b, \tilde{X}_c, \tilde{X}_d$. Here, the labels $a...d$ are defined as sets of flavor indices:

$$a \equiv xx|xx \quad b \equiv xx|yy \quad c \equiv xy|xy \quad d \equiv xy|yx. \tag{18}$$

All other combinations of flavors are either zero (e.g. the types $xx|yz$), or may be transformed into the ones above via global $SO(3)$ rotations (e.g. $zz|xx \to xx|yy$). The $d$ type channels need to be defined since the first and last two indices may no longer be permuted separately for $\tilde{X}$ type vertices. This finally allows us to write Eq. (16) as:

$$\dot{\Gamma}^{\Lambda\,2L}_{a\,ij}(s,t,u) = Y^\Lambda_{a\,ij}(s,t,u) - \tilde{Y}^\Lambda_{a\,ij}(t,s,u) + \tilde{Y}^\Lambda_{a\,ij}(u,s,t) \tag{19a}$$

$$\dot{\Gamma}^{\Lambda\,2L}_{b\,ij}(s,t,u) = Y^\Lambda_{b\,ij}(s,t,u) - \tilde{Y}^\Lambda_{c\,ij}(t,s,u) + \tilde{Y}^\Lambda_{c\,ij}(u,s,t) \tag{19b}$$

$$\dot{\Gamma}^{\Lambda\,2L}_{c\,ij}(s,t,u) = Y^\Lambda_{c\,ij}(s,t,u) - \tilde{Y}^\Lambda_{b\,ij}(t,s,u) + \tilde{Y}^\Lambda_{d\,ij}(u,s,t) \tag{19c}$$

where $\tilde{Y}^\Lambda_{d\,ii}(s,t,u) = -\tilde{Y}^\Lambda_{c\,ii}(s,u,t) = -Y^\Lambda_{c\,ii}(s,u,t)$ and the definitions of $Y_a$ etc. are given in Appendix B.2.

## B.2 Symmetries

For the numerical implementation of the $X$, $\tilde{X}$, $Y$ and $\tilde{Y}$-terms, symmetries of the transfer frequencies $s$, $t$ and $u$ are crucial. In analogy to Ref. [13], the identities summarized in Table 1 can be proven.

| Operation | $X^{\Lambda}_{\mu,\,ij}(s,t,u)$ | $\tilde{X}^{\Lambda}_{\mu,\,ij}(s,t,u)$ |
|---|---|---|
| $1 \leftrightarrow 2$ | $X_{a/b}(s,t,u)$ $\leftrightarrow -X_{a/b}(s,u,t)$ | not allowed |
| $T \circ (1,3) \leftrightarrow (2,4)$ | $s \leftrightarrow -s$ | $s \leftrightarrow -s$ , $i \leftrightarrow j$ |
| $T \circ (1,2) \leftrightarrow (3,4)$ | $t \leftrightarrow -t$ , $i \leftrightarrow j$ | $t \leftrightarrow -t$ |
| $T \circ (1,2) \leftrightarrow (4,3)$ | $u \leftrightarrow -u$ , $i \leftrightarrow j$ | $u \leftrightarrow -u$ , $i \leftrightarrow j$ |

Table 1: Transformations of the frequency arguments under time reversal $T$ and specific permutations of indices in $X^{\Lambda\,ij}_{1,2|3,4}$ and $\tilde{X}^{\Lambda\,ij}_{1,2|3,4}$. The exchange $1 \leftrightarrow 2$ would change $X_c$ to the form $X_{xyyx}$ and $\tilde{X}$ to $X_{ji|ij}$. Hence, the resulting symmetries take the slightly different form in Eq. (23). Equivalent relations hold for $X^{\Lambda} \to Y^{\Lambda}$ and $\tilde{X}^{\Lambda} \to \tilde{Y}^{\Lambda}$.

Finally, we prove an identity which eliminates the need of implementing a flow equation for the $d$-type-bubble functions. With the starting equation Eq. (20a) being a result of global $SO(3)$ symmetry as proven in Ref. [13] we have:

$$\Gamma^{\Lambda,\,\mu}_{xxxx} = \Gamma^{\Lambda,\,\mu}_{xxyy} + \Gamma^{\Lambda,\,\mu}_{xyxy} + \Gamma^{\Lambda,\,\mu}_{xyyx} \tag{20a}$$

$$\Rightarrow X^{\Lambda,\,\mu}_{xx|xx} = X^{\Lambda,\,\mu}_{xx|yy} + X^{\Lambda,\,\mu}_{xy|xy} + X^{\Lambda,\,\mu}_{xy|yx} \tag{20b}$$

$$\Rightarrow Y^{\Lambda,\,\mu}_{xx|xx} = Y^{\Lambda,\,\mu}_{xx|yy} + Y^{\Lambda,\,\mu}_{xy|xy} + Y^{\Lambda,\,\mu}_{xy|yx}, \tag{20c}$$

where $\mu \equiv (i_1, i_2, i_3, i_4, \omega_1, \omega_2, \omega_3, \omega_4)$ refers to an arbitrary fixed set of site and frequencies, noting that no use of permutation symmetry is made in the following. To demonstrate that Eqs. (20b) and (20c) follow from Eq. (20a), the latter is inserted into the definitions of the one-loop and two-loop channel functions Eqs. (2d) and (16). Using that propagators are diagonal and computing the flavor summation first before any site or frequency parametrization is applied, we obtain

$$X^{\Lambda,\,\mu}_{\alpha_1\alpha_2|\alpha_3\alpha_4} \sim \sum_{\beta_1,\beta_3} \Gamma^{\Lambda,\,\nu}_{\alpha_1\alpha_2|\beta_3\beta_1} \Gamma^{\Lambda,\,\rho}_{\beta_1\beta_3|\alpha_3\alpha_4}. \tag{21}$$

Here, for convenience of notation, the propagators are kept only implicitly. After inserting external flavor labels on the left, the summation can be carried out so that

$$X^{\Lambda,\,\mu}_{xx|xx} \sim \Gamma^{\Lambda,\,\nu}_{xxxx}\Gamma^{\Lambda,\,\rho}_{xxxx} + 2\Gamma^{\Lambda,\,\nu}_{xxyy}\Gamma^{\Lambda,\,\rho}_{xxyy}$$

$$X^{\Lambda,\,\mu}_{xx|yy} \sim \Gamma^{\Lambda,\,\nu}_{xxyy}\Gamma^{\Lambda,\,\rho}_{xxxx} + \Gamma^{\Lambda,\,\nu}_{xxxx}\Gamma^{\Lambda,\,\rho}_{xxyy} + \Gamma^{\Lambda,\,\nu}_{xxyy}\Gamma^{\Lambda,\,\rho}_{xxyy}$$

$$X^{\Lambda,\,\mu}_{xy|xy} \sim \Gamma^{\Lambda,\,\nu}_{xyyx}\Gamma^{\Lambda,\,\rho}_{xyxy} + \Gamma^{\Lambda,\,\nu}_{xyxy}\Gamma^{\Lambda,\,\rho}_{xyyx}$$

$$X^{\Lambda,\,\mu}_{xy|yx} \sim \Gamma^{\Lambda,\,\nu}_{xyxy}\Gamma^{\Lambda,\,\rho}_{xyxy} + \Gamma^{\Lambda,\,\nu}_{xyyx}\Gamma^{\Lambda,\,\rho}_{xyyx}. \tag{22}$$

Equation (20b) may then be proven by inserting these expressions into it and subsequently using Eq. (20a) on all occurring instances of $\Gamma^{\nu}$ and $\Gamma^{\rho}$ to verify the equivalence of the left and right

hand side. This procedure may be repeated for the definition of the two-loop contributions to finally prove Eq. (20c).

As a result of this symmetry, we do not need to compute $X^{\Lambda}_{c\,ij}(s,u,t)$ and $Y^{\Lambda}_{c\,ij}(s,u,t)$ for $t > u$ and in particular no flow equation is required for $\tilde{X}^{\Lambda}_d$ and $\tilde{Y}^{\Lambda}_d$ since Eq. (20b) can be written as

$$X^{\Lambda}_{c\,ij}(s,u,t) = \left(-X^{\Lambda}_{a\,ij} + X^{\Lambda}_{b\,ij} + X^{\Lambda}_{c\,ij}\right)(s,t,u) \tag{23a}$$

$$\tilde{X}^{\Lambda}_{d\,ij}(s,t,u) = \left(\tilde{X}^{\Lambda}_{a\,ij} - \tilde{X}^{\Lambda}_{b\,ij} - \tilde{X}^{\Lambda}_{c\,ij}\right)(s,t,u). \tag{23b}$$

**Explicit parametrization of bubble functions**

Using the one-loop bubble functions $X$ and $\tilde{X}$ from Ref. [13], the two-loop bubble functions can be given explicity. In the equations below, the propagator is $i\mathcal{G}^{\Lambda}_i(\omega) = \frac{\omega}{\omega^2 + \omega\gamma_i(\omega) + \Lambda^2}$, with $\gamma_i(\omega) = i\Sigma_i(\omega)$. While the site index is kept here for generality, it can be dropped in the case of lattices consisting of equivalent sites only.

$$Y^{\Lambda}_{a\,ij} = -T\sum_{\omega}\sum_{k}\mathcal{G}_k(\omega)\mathcal{G}_k(s+\omega)\Big[$$
$$\left(\Gamma^{\Lambda}_{a,\,ki}(s,\omega+\omega_1,\omega+\omega_2)\tilde{X}^{\Lambda}_{a,\,kj}(\omega-\omega_4,s,\omega-\omega_3)\right.$$
$$\left.+\Gamma^{\Lambda}_{a,\,kj}(s,\omega-\omega_3,\omega-\omega_4)\tilde{X}^{\Lambda}_{a,\,ki}(\omega+\omega_2,s,\omega+\omega_1)\right)$$
$$+2(\Gamma^{\Lambda}_a \to \Gamma^{\Lambda}_b, \tilde{X}^{\Lambda}_a \to \tilde{X}^{\Lambda}_c)\Big] \tag{24a}$$

$$Y^{\Lambda}_{b\,ij} = -T\sum_{\omega}\sum_{k}\mathcal{G}_k(\omega)\mathcal{G}_k(s+\omega)\Big[$$
$$\left(\Gamma^{\Lambda}_{b,\,ki}(s,\omega+\omega_1,\omega+\omega_2)\tilde{X}^{\Lambda}_{a,\,kj}(\omega-\omega_4,s,\omega-\omega_3)\right.$$
$$\left.+\Gamma^{\Lambda}_{b,\,kj}(s,\omega-\omega_3,\omega-\omega_4)\tilde{X}^{\Lambda}_{a,\,ki}(\omega+\omega_2,s,\omega+\omega_1)\right)$$
$$+(\Gamma^{\Lambda}_b \to \Gamma^{\Lambda}_a, \tilde{X}^{\Lambda}_a \to \tilde{X}^{\Lambda}_c) + (\tilde{X}^{\Lambda}_a \to \tilde{X}^{\Lambda}_c)\Big] \tag{24b}$$

$$Y^{\Lambda}_{c\,ij} = -T\sum_{\omega}\sum_{k}\mathcal{G}_k(\omega)\mathcal{G}_k(s+\omega)\Big[$$
$$\Gamma^{\Lambda}_{c,\,ki}(s,\omega+\omega_2,\omega+\omega_1)\tilde{X}^{\Lambda}_{b,\,kj}(\omega-\omega_4,s,\omega-\omega_3)$$
$$+\Gamma^{\Lambda}_{c,\,kj}(s,\omega-\omega_4,\omega-\omega_3)\tilde{X}^{\Lambda}_{b,\,ki}(\omega+\omega_2,s,\omega+\omega_1)$$
$$-\Gamma^{\Lambda}_{c,\,ki}(s,\omega+\omega_1,\omega+\omega_2)\tilde{X}^{\Lambda}_{d,\,kj}(\omega-\omega_4,s,\omega-\omega_3)$$
$$-\Gamma^{\Lambda}_{c,\,kj}(s,\omega-\omega_3,\omega-\omega_4)\tilde{X}^{\Lambda}_{d,\,ki}(\omega+\omega_2,s,\omega+\omega_1)\Big] \tag{24c}$$

$$\tilde{Y}^{\Lambda}_{a\,ij} = -T\sum_{\omega}\mathcal{G}_i(\omega)\mathcal{G}_j(s+\omega)\Big[$$
$$\left(\Gamma^{\Lambda}_{a,\,ji}(\omega-\omega_3,s,\omega-\omega_4)\tilde{X}^{\Lambda}_{a,\,ji}(\omega+\omega_2,\omega+\omega_1,s)\right.$$
$$\left.+\Gamma^{\Lambda}_{a,\,ji}(\omega+\omega_1,s,\omega+\omega_2)\tilde{X}^{\Lambda}_{a,\,ji}(\omega-\omega_4,\omega-\omega_3,s)\right)$$
$$+2(\Gamma^{\Lambda}_a \to \Gamma^{\Lambda}_c, \tilde{X}^{\Lambda}_a \to \tilde{X}^{\Lambda}_d)\Big]$$
$$-T\sum_{\omega}\mathcal{G}_j(\omega)\mathcal{G}_i(s+\omega)\Big[$$
$$\left(\Gamma^{\Lambda}_{a,\,ij}(\omega+\omega_2,s,\omega+\omega_1)X^{\Lambda}_{a,\,ij}(\omega-\omega_4,s,\omega-\omega_3)\right.$$
$$\left.+\Gamma^{\Lambda}_{a,\,ij}(\omega-\omega_4,s,\omega-\omega_3)X^{\Lambda}_{a,\,ij}(\omega+\omega_2,s,\omega+\omega_1)\right)$$
$$+2(\Gamma^{\Lambda}_a \to \Gamma^{\Lambda}_c, X^{\Lambda}_a \to X^{\Lambda}_c)\Big] \tag{25a}$$

$$\tilde{Y}_{b\ ij}^{\Lambda} = -T\sum_{\omega}\mathcal{G}_i(\omega)\mathcal{G}_j(s+\omega)\Big[$$

$$\Big(\Gamma_{c,\ ji}^{\Lambda}(\omega+\omega_1,s,\omega+\omega_2)\tilde{X}_{a,\ ji}^{\Lambda}(\omega-\omega_4,\omega-\omega_3,s)$$
$$+\Gamma_{c,\ ji}^{\Lambda}(\omega-\omega_3,s,\omega-\omega_4)\tilde{X}_{a,\ ji}^{\Lambda}(\omega+\omega_2,\omega+\omega_1,s)\Big)$$
$$+(\Gamma_c^{\Lambda}\rightarrow\Gamma_a^{\Lambda},\tilde{X}_a^{\Lambda}\rightarrow\tilde{X}_d^{\Lambda})+(\tilde{X}_a^{\Lambda}\rightarrow\tilde{X}_d^{\Lambda})\Big]$$
$$-T\sum_{\omega}\mathcal{G}_j(\omega)\mathcal{G}_i(s+\omega)\Big[$$

$$\Big(\Gamma_{a,\ ij}^{\Lambda}(\omega+\omega_2,s,\omega+\omega_1)X_{c,\ ij}^{\Lambda}(\omega-\omega_4,s,\omega-\omega_3)$$
$$+\Gamma_{a,\ ij}^{\Lambda}(\omega-\omega_4,s,\omega-\omega_3)X_{c,\ ij}^{\Lambda}(\omega+\omega_2,s,\omega+\omega_1)\Big)$$
$$+(\Gamma_a^{\Lambda}\rightarrow\Gamma_c^{\Lambda},X_c^{\Lambda}\rightarrow X_a^{\Lambda})+(\Gamma_a^{\Lambda}\rightarrow\Gamma_c^{\Lambda})\Big] \tag{25b}$$

$$\tilde{Y}_{c\ ij}^{\Lambda} = T\sum_{\omega}\mathcal{G}_i(\omega)\mathcal{G}_j(s+\omega)\Big[$$

$$\Big(\Gamma_{c,\ ji}^{\Lambda}(\omega+\omega_1,\omega+\omega_2,s)\tilde{X}_{b,\ ji}^{\Lambda}(\omega-\omega_4,\omega-\omega_3,s)$$
$$+\Gamma_{c,\ ji}^{\Lambda}(\omega-\omega_3,\omega-\omega_4,s)\tilde{X}_{b,\ ji}^{\Lambda}(\omega+\omega_2,\omega+\omega_1,s)\Big)$$
$$+(\Gamma_c^{\Lambda}\rightarrow\Gamma_b^{\Lambda},\tilde{X}_b^{\Lambda}\rightarrow\tilde{X}_c^{\Lambda})\Big]$$
$$-T\sum_{\omega}\mathcal{G}_j(\omega)\mathcal{G}_i(s+\omega)\Big[$$

$$\Big(\Gamma_{b,\ ij}^{\Lambda}(\omega+\omega_2,\omega+\omega_1,s)X_{b,\ ij}^{\Lambda}(\omega-\omega_4,\omega-\omega_3,s)$$
$$+\Gamma_{b,\ ij}^{\Lambda}(\omega-\omega_4,\omega-\omega_3,s)X_{b,\ ij}^{\Lambda}(\omega+\omega_2,\omega+\omega_1,s)\Big)$$
$$+(\Gamma_b^{\Lambda}\rightarrow\Gamma_c^{\Lambda},X_b^{\Lambda}\rightarrow X_c^{\Lambda})\Big] \tag{25c}$$

and as a consequence of Eq. (20c)

$$\tilde{Y}_{d\ ij}^{\Lambda} = \tilde{Y}_{a,\ ij}^{\Lambda}(s,t,u) - \tilde{Y}_{b,\ ij}^{\Lambda}(s,t,u) - \tilde{Y}_{c,\ ij}^{\Lambda}(s,t,u). \tag{25d}$$

## C Details on the numerical implementation

The solution of the flow equations amounts to the numerical integration of a large system of coupled ordinary differential equations (ODE's). The initial conditions are given as

$$\Gamma_{c\ ij}^{\Lambda_0}(s,t,u) = -J_{ij}$$
$$\gamma_i^{\Lambda_0}(\omega) = \Gamma_{a\ ij}^{\Lambda_0}(s,t,u) = \Gamma_{b\ ij}^{\Lambda_0}(s,t,u) = 0, \tag{26}$$

with $\Lambda_0$ at least two orders of magnitude above the largest exchange coupling. To obtain a finite system of equations, only the first $N_\omega$ non-negative Matsubara frequencies are considered (negative frequencies are related by symmetries). Matsubara sums over $i\omega_n$ are truncated for $|n| > N_w$. The error made in this approximation is controlled since the contribution of large frequencies is typically small due to the vanishing propagator $\mathcal{G}(i\omega_n) \sim 1/i\omega_n$. For four-point vertices, we must pay special attention to the fact that combinations of bosonic Matsubara integers $n_s, n_t, n_u$ are (un-)physical if their sum $n_s + n_t + n_u$ is odd (even) [13]. Vertices with unphysical frequency arguments will never appear in flow equations and are thus not computed. If one or more Matsubara integers are greater or equal to $N_\omega$, the vertex is approximated by setting the associated index to either $N_\omega - 1$ or $N_\omega - 2$ such that $n_s + n_t + n_u$ is odd. This avoids the introduction of

errors at the boundaries of our frequency range. For the same reason, we also refrain from the alternative of interpolating between frequencies and instead raise the number of positive frequencies until convergence is reached. Good results are typically obtained at $N_\omega = 32$, particularly, for temperatures $T \gtrsim 0.5$. For the lowest temperature treated, $T_{\min} = 0.2$, we found full convergence of the structure factor below $N_\omega = 64$, while convergence of the energy per site required a higher number of $N_\omega = 96$. At $T = 0.2$, the latter value corresponds to a maximum bosonic frequency of $\approx 120 J_1$, more than two orders of magnitude larger than the relevant energy scale.

Regarding the real space cutoff discussed in the main text, we report no significant dependence on the particular choice of the cutoff. If the maximum vertex length is defined by the number of nearest-neighbor bonds instead of an (isotropic) distance $L$, the same scaling behaviour is observed.

Numerically, the flow equations were solved using adaptive, error-controlled methods provided in the Julia package "DifferentialEquations.jl" [55]. To allow for accurate numerical derivatives of the free energy, a relative tolerance $\sim 10^{-8}$ is required in which case the Dormand-Prince(5) method was found to be most efficient.

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
