# Peer review of "Quantitative functional renormalization for three-dimensional quantum Heisenberg models"

_SciPost Physics_

## Round 1 · Referee Report · Anonymous · 2022-2-10

Strengths
(1) The manuscript applies a recent methodical development within the functional RG -- a powerful quantum many-body framework -- to a collection of interesting (unsolved) spin models.
(2) The new method is first benchmarked at the example of a well-established three-dimensional model, i.e. on the simple cubic lattice.
(3) The method is then shown to produce a variety of non-trivial results for frustrated three-dimensional frustrated systems, which are generally difficult to access with other methods.
(4) The method is very promising for future applications.
Weaknesses
(1) The effect of the "two-loop" corrections is not discussed for the simple cubic lattice, which would be an important convergence check for the benchmarks in Sec. 3.
(2) The discussion in Sec. 3 leaves the open question whether, the data would also be compatible with the mean-field exponents \nu=1/2 and \eta=0 for a slightly different Tc.
(3) The merits of taking into account the "two-loop" corrections are unclear, in fact, they appear somewhat accidental.
Report
The manuscript is about the application of a recently developed extension of the functional RG approach to quantum spin systems where the spin degrees of freedom are represented in terms of auxiliary fermions. The present work specifically employs a Majorana representation, which has been argued in previous work (of the same authors) to provide a series of advantages as compared to a decomposition into complex fermions. A strength of the approach is its flexibility and its applicability to a broad range of unsolved problems, including the case of three-dimensional spin systems, both unfrustrated and frustrated, as is the subject of this work. Therefore, the manuscript can be seen as laying the groundwork in this existing research field providing the clear potential for significant follow-up work.
The manuscript is very well written and structured and it is a pleasant read. The considered problem is well introduced and properly motivated. Technicalities of the method are introduced appropriately in the main text and ample details are given in the appendices. References and the concluding discussion are appropriate, but could be slightly revised according to the comments and questions that I will present in the following (and which are listed, again, with the same numbering in the section on the requested changes):
(1) In Sec. 3, the authors present present benchmark calculations for the simple cubic lattice without frustration and compare their results to previous QMC results. The benchmark critical exponents for the Heisenberg universality class they cite seem a bit outdated. While this doesn't really matter for the precision that is achieved in the present work, very recent conformal bootstrap calculations could be referred to at this point, e.g., PRD 104, 105013 (2021). Also, it could be interesting to refer to related FRG work on Heisenberg universality, see PRB 97, 075129.
(2) I'm not sure whether I find the compatibility check of the critical exponents convincing. More concretely: if I would assume that the method reproduces the mean-field exponents nu=1/2 and eta=0, would I also get a reasonable scaling collapse and a decent estimate for the critical temperature? My naive expectation would be that the fermionic RG equations on the 1-loop level should reproduce mean-field exponents because they do not take into account order-parameter fluctuations. Since at least the anomalous dimension is very small anyway, this would also explain the apparent "compatibility" with the quantitative value. It would be good, if the authors could comment on that.
(3) Can the critical temperature not be determined in a more direct way? This would facilitate also a more straightforward calculation of the critical exponents.
(4) The comparisons for the simple cubic case are exclusively done for what the authors refer to as the "1-loop" truncation and they argue that their results are quantitatively accurate. For that statement to be convincing, it would be essential that these results converge with increasing loop order at least if the considered loop expansion is systematic for a finite amount of loops otherwise the agreement would seem somewhat accidental. This seems to be particularly important as in the next section, they make a case for taking into account "2-loop" contributions. With this in mind, I would like to strongly encourage the authors to discuss the effect of 2-loop corrections also for the case in Sec. 3.
(5) Towards the end of section 3, the authors suggest to study first-order transitions with the presented method. How would the method manage to distinguish a first- from a second-order transition? I think this should be properly explained otherwise the promise to investigate this in future work appears a bit speculative.
(6) At the end of section 3, the authors claim that an advantage of PMFRG over PFFRG is that they can consider finite temperatures. To my understanding, this is probably not an issue of the fermion representation, but an issue of the regulator choice, i.e. with a different regulator scheme it should be also possible for PFFRG to handle finite temperatures (maybe if the particle number constraint is implemented differently).
(7) In section 4 it is discussed that the 2-loop correction induces a 10% shift of the numerical results. Can this also be expected for the simple cubic lattice? How does this affect the quantitative results?
(8) In Section 4, the authors also mention, again, that "temperature below J \sim 0.2J_1 is currently not accessible". While this is not a particularly low temperature, especially when it comes to correlated effects in frustrated systems, it would be very interesting to know what needs to be improved in the future to get to smaller temperatures within the presented method. What does this depend on?
(9) The discussion of the 2-loop corrections in Sec. 5 is interesting but if I'm willing to accept the validity of the 1-loop results in Sec. 3, I'm quite sceptical about the 2-loop, in particular, since the authors provide an argument for an even-odd alternating behavior with the loop order. Overall this does not make a convincing case for the loop expansion in my opinion. I understand that the infinite-loop case recovers the parquet approximation diagrammatically, but in view of the presented results, I'm wondering whether the finite-loop truncation provides a reasonably convergent scheme or whether it is somehow flawed. I think this should be clarified, otherwise I wouldn't find the two-loop results very convincing.
Requested changes
(1) Optionally consider updated references for the Heisenberg universality class.
(2) Comment on the mean-field exponents in Sec. 3 and calculate Tc with mean-field exponents for comparison if possible.
(3) Explain whether the critical temperature can be extracted more directly.
(4) Comment on the 2-loop corrections for the case of Sec. 3 and ideally provide calculations of Tc and the exponents at two-loop order.
(5) Explain how to distinguish first- from second-order transitions with the presented method at the end of section 3 or remove that claim, if it is too speculative.
(6) At the end of section 3, please comment on the issue of finite temperature when comparing PFFRG and PMFRG.
(7) Comment on the convergence of results with increasing loop order in the discussion in Sec. 4.
(8) Explain what sets the limit for achieving lower temperatures and why T \sim 0.2 J_1 seems to be the limit.
(9) Discuss systematics behind the finite-loop truncation of the multi-loop approach.
(10) The introduction and conclusion has to be revised appropriately, in order to reflect the revisions made in sections 3,4, and 5.
Anonymous on 2022-02-22 [id 2232]
We thank the referee for pointing out several important questions which we explicitly address below. Furthermore, we have also added clarifications regarding these points in the attached manuscript (blue font). We will refer to each of the referee's remarks individually according to the numbers provided in the list of changes.
Concerning the critical exponents, we thank the referee for pointing us to additional relevant references in (1), and we cited them in our revised manuscript.
Regarding critical temperatures and mean-field exponents [points (2) and (3)], we emphasize that in our procedure to verify the consistency with universal scaling laws, a careful separation is made between the extraction of the critical temperature and the scaling collapse, showing consistency with the critical exponents. In Fig. 1 (b), the critical temperature is determined under no assumption about the values of the critical exponents. T_c is determined unambiguously by identifying the one temperature which results in a pure power-law relation between system size L and the susceptibility. Only after the value of T_c is established this way, it becomes possible to verify the compatibility with classical critical exponents via the scaling collapse in Fig.1 (c). We stress that allowing for different values of the critical temperature would generically lead to an under-determined set of parameters T_c, \eta, \nu in which a scaling collapse would not yield any physical information.
To clarify this important point, we have added a panel (d) demonstrating the poor quality of the collapse plot using mean-field exponents. However, we do agree with the referee that the level of accuracy does not exclude a vanishing anomalous dimension. We emphasize that the strength of the PMFRG lies within its capability to treat microscopic models of frustrated quantum magnets, and is not meant to compete with established high-precision methods to extract critical exponents from effective field theories.
Further, in response to the questions about second and higher loop corrections [points (4) and (7)], we referenced in our manuscript the observations that we discuss in more detail in section 5. Due to the inherent complication of treating a purely interacting model, all diagrammatic approximations should be gauged mainly by benchmarking their predictive power in various applications. Most importantly, we found that the inclusion of two-loop contributions has different effects in the magnetically ordered and paramagnetic cases. Explicitly, we mention that while they are observed to improve numerical accuracy in the pyrochlore case, they suppress magnetic order, rendering the application of finite-size scaling impossible, so that in particular no quantitative results regarding Tc can be found. We discuss the underlying diagrammtic reason for this and give an outlook on how this problem can possibly be resolved upon including higher loop orders, but leave this much more involved investigation for future work.
As the referee points out, the PMFRG currently suffers from unphysical behaviour for $T<0.2J$, which, although shown to be in line with other state-of the art approaches such as DMRG, is still too high to resolve the extremely low energy scales that are important in many quantum effects. In answer to the question on the minimum achievable temperature [points (8) and (9)] we thus motivate our inclusion of two-loop contributions by the fact that they include higher order diagrams, making the FRG scheme exact up to fourth order in the interaction strength. Physically, the only approximation made in the PMFRG is the truncation of flow equations. The multi-loop expansion, as the referee correctly mentions, is motivated from its convergence to the parquet approximation and as of now the only systematic way to improve the accuracy of the FRG formalism. While it is our belief that this will improve our presented scheme to enable lower temperatures and recovers the correct scaling behaviour found in the one-loop scheme, this remains to be seen, since due to the purely interacting nature of the Heisenberg model, any approximation neglecting certain classes of diagrams may prove inaccurate for certain applications.
Concerning the distinction between first and second order transitions [point (5)], we have clarified in our manuscript that even though the susceptibility displays a strong growth in the suspected first-order transition regime, the scaling behaviour upon increasing L does not correspond to the finite-size scaling laws. While a phase transition to an ordered phase is trivially expected in the limit of infinite J_2, our results seem to indicate that this phase transition is not of second order which is in agreement with the observations made in the classical model. While at the current stage, we cannot directly identify a first order phase transition with the same rigor, it is nonetheless an interesting question to study in the future that should be mentioned in the outlook.
Finally, the referee asks for clarification about the comparison between PFFRG and PMFRG at finite temperature [point (6)]. While in the text, we mostly refer to our previous work (Ref. 13) for a detailed discussion, we point out in our present work that the inapplicability of the PFFRG at finite temperature stems from the presence of unphysical eigenstates in the complex fermionic spin representation. The pseudo-Majorana representation used here, on the other side introduces no unphysical states (but only copies of physical ones) and, hence, does not suffer from this limitation.
Corresponding to these changes, we have revised our introduction and conclusion as proposed by the referee [point (10)].
Attachment:
Quantitative_PMFRG_changes.pdf

---

## Round 1 · Referee Report · Anonymous · 2022-3-16

Strengths
thorough two-loop analysis with the PMFRG applied to 3D Heisenberg models
Weaknesses
- justification of two-loop in view of multiloop approximation ("quantitative" in the title is questionable), see below
- possible overlap with Ref. 16
Report
In the present manuscript, the authors present a two-loop PMFRG study applied to 3D unfrustrated and frustrated Heisenberg models no the simple cubic and pyrochlore lattices. It turns out, that for the specific models considered, the two-loop results for the susceptibilities, the ordering temperature, as well as the critical exponent and the anomalous dimension are accurately described.
The presentation is clear and the provided analysis scientifically sound. The presented results are interesting for improving on the standard one-loop FRG mainly used up to now.
However, I recommend to address a few points before publication and modify the manuscript accordingly, in particular concerning the presentation in the title and abstract.
Neglecting higher loop orders beyond the presented two-loop computations is not well justified, besides the substantial quantitative improvement with respect to one loop. There is mentioned only an increased numerical effort for the computation of higher loop orders, which is referred to future work. Their effect has been extensively discussed in Refs. 16 an 17 (and also in SciPost Phys. 6, 009 (2019) and Phys. Rev. Research 2, 033372 for fermionic models, which could be included among the multiloop FRG literature), where their relevance for the loop convergence has been illustrated. In the abstract it is mentioned that "extensions towards higher loop orders seem to systematically improve the approach for magnetically disordered systems", but their investigation is missing here. This statement appears also contrasting the 'quantitative' in the title which I feel not appropriate and rather misleading. Similarly, the concluding statement "Overall, the pseudo Majorana functional renormalization group is established as a powerful many-body technique in quantum magnetism with a wealth of possible future applications." rather fits Refs. 16 an 17 where this approach has been introduced.
Moreover, in Ref. 16, the PMFRG has been applied also to antiferromagnetic Heisenberg models on the pyrochlore and simple cubic lattices for a benchmark study. It should be clearly stated in which respect the present analysis goes beyond that study.
Requested changes
see report

---

## Editorial Decision

unknown